# Potential Risk of Overtreatment in Patients with Type 2 Diabetes Aged 75 Years or Older: Data from a Population Database in Catalonia, Spain

**DOI:** 10.3390/jcm11175134

**Published:** 2022-08-31

**Authors:** Manel Mata-Cases, Didac Mauricio, Jordi Real, Bogdan Vlacho, Laura Romera-Liebana, Núria Molist-Brunet, Marta Cedenilla, Josep Franch-Nadal

**Affiliations:** 1DAP-Cat Group, Unitat de Suport a la Recerca Barcelona Ciutat, Institut Universitari d’Investigació en Atenció Primària Jordi Gol (IDIAP Jordi Gol), 08025 Barcelona, Spain; 2CIBER of Diabetes and Associated Metabolic Diseases (CIBERDEM CB15/00071), Instituto de Salud Carlos III (ISCIII), 28029 Madrid, Spain; 3Primary Health Care Center La Mina, Gerència d’Àmbit d’Atenció Primària Barcelona Ciutat, Institut Català de la Salut, Sant Adrià de Besòs, 08930 Barcelona, Spain; 4Department of Endocrinology & Nutrition, Hospital de la Santa Creu i Sant Pau, 08025 Barcelona, Spain; 5Departament of Medicine, University of Vic-Central University of Catalonia (UVIC-UCC), 08500 Vic, Spain; 6Primary Healthcare Centre Raval Nord, Institut Català de la Salut, 08001 Barcelona, Spain; 7Hospital Universitari de la Santa Creu de Vic, 08500 Vic, Spain; 8Central Catalonia Chronicity Research Group (C3RG), Centre for Health and Social Care Research (CESS), University of Vic-Central University of Catalonia (UVIC-UCC), 08500 Vic, Spain; 9Medical Affairs Department, MSD Spain, 28027 Madrid, Spain; 10Primary Health Care Center Raval Sud, Gerència d’Àmbit d’Atenció Primària Barcelona Ciutat, Institut Català de la Salut, 08001 Barcelona, Spain

**Keywords:** type 2 diabetes, treatment, overtreatment, older diabetes adults

## Abstract

Aim: To assess the potential risk of overtreatment in patients with type 2 diabetes (T2DM) aged 75 years or older in primary care. Methods: Electronic health records retrieved from the SIDIAP database (Catalonia, Spain) in 2016. Variables: age, gender, body mass index, registered hypoglycemia, last HbA1c and glomerular filtration rates, and prescriptions for antidiabetic drugs. Potential overtreatment was defined as having HbA1c < 7% or HbA1c < 6.5% in older patients treated with insulin, sulfonylureas, or glinides. Results: From a total population of 138,374 T2DM patients aged 75 years or older, 123,515 had at least one HbA1c available. An HbA1c below 7.0% was present in 59.1% of patients, and below 6.5% in 37.7%. Overall, 23.0% of patients were treated with insulin, 17.8% with sulfonylureas, and 6.6% with glinides. Potential overtreatment (HbA1c < 7%) was suspected in 26.6% of patients treated with any high-risk drug, 47.8% with sulfonylureas, 43.5% with glinides, and 28.1% with insulin. Using the threshold of HbA1c < 6.5%, these figures were: 21.6%, 24.4%, 17.9%, and 12.3%, respectively. Conclusion: One in four older adults with T2DM treated with antidiabetic drugs associated with a high risk of hypoglycemia might be at risk of overtreatment. This risk is higher in those treated with sulfonylureas or glinides than with insulin.

## 1. Introduction

The rising prevalence of type 2 diabetes mellitus (T2DM) in most parts of the world is likely to increase the incidence of the chronic complications associated with the disease [1]. This calls for the improved treatment of hyperglycemia and other risk factors associated with T2DM to lower the risk of both micro- and macrovascular complications, as all international and national consensus documents and guidelines propose [2,3,4]. However, in older patients with multimorbidity or those who are frail, the benefits of tight glycemic control decline and the risk and burden of antidiabetic treatments increase [5,6,7]. Moreover, observational studies of people with T2DM who are older or have high clinical complexity have found an association between tight blood glucose control (HbA1c < 7% (53 mmol/mol)) and higher risk of falls, severe hypoglycemia, emergency department visits, hospitalizations, and death [7,8,9,10,11]. Trying to achieve tight glycemic levels could be detrimental in older patients, particularly those with complications and serious comorbidities, because of their high likelihood of severe hypoglycemia [5,6,7]. Such episodes in older adults have implications for both short-term (e.g., risk of falls, accidents, hospitalizations, and death) and long-term negative outcomes (e.g., lower quality of life, decreased cognitive function, and increased risk of cardiovascular mortality) [7]. Some real-world population studies on older adults with T2DM have consistently shown that, despite the fact that they are well controlled (HbA1c < 7%), they are treated with antidiabetic drugs associated with a high risk of hypoglycemia (insulin, sulfonylureas, and glinides) and, thus, they might be at risk of overtreatment [12,13,14,15,16,17,18].

Primary healthcare professionals face the challenge of implementing increasingly complex treatment algorithms and overcoming therapeutic inertia in daily practice, which results not only in delayed treatment intensification but also in delayed treatment deintensification among the old and frail [19]. Deintensification has been incorporated into some therapeutic guidelines [20,21,22] and has been shown to be safe and effective in a metanalysis [23]. However, deintensification is relatively infrequent in clinical practice [24,25,26]. The failure to deintensify therapy raises safety concerns and may contribute to overtreatment; in turn, it may lead to avoidable direct and indirect health costs [6].

Several reports on the management and treatment of patients with T2DM in Catalonia (Spain), based on the SIDIAP population database, have consistently shown that older adults have better control than other age subgroups [27,28,29,30]. For instance, 59.2% of T2DM patients over 75 had HbA1c < 7% and 37.7% HbA1c < 6.5% in 2016 [30], which leads us to suspect the existence of potential overtreatment. We undertook this study to specifically analyze the potential risk of overtreatment defined as an HbA1c < 7% or an HbA1c < 6.5% in patients aged 75 years or older treated with drugs associated with a higher risk of hypoglycemia (insulin, sulfonylureas, or glinides). Moreover, we evaluated the use of high-risk antidiabetic drugs, potential overtreatment, and the presence of hypoglycemia in cardiovascular and renal comorbidities.

## 2. Methods

This was a cross-sectional study using the SIDIAP database (Information System for the Development of Research in Primary Care) (www.sidiap.org, accessed on 7 August 2022) [28], which contains pseudo-anonymized longitudinal patient information obtained from the electronic health records (EHR) of patients attended by 286 primary care teams at the Institut Català de la Salut (Catalan Health Institute), which covered 74% of the total population in Catalonia in 2016 [29]. In brief, SIDIAP contains sociodemographic characteristics, morbidity (International Classification of Diseases, Tenth Revision; ICD-10), clinical and lifestyle variables, specialist referrals, laboratory tests, and treatments (prescription and pharmacy invoicing data) since 2006. All primary care health centers use the same EHR, called the Primary Care Clinical Station (in Catalan: Estació Clínica d’Atenció Primària, ECAP). Catalonia is a Mediterranean region in northeastern Spain, with a public health system in which every citizen is registered with a general practitioner and a nurse in a publicly funded primary care health center. Healthcare and all diagnostic procedures are free of charge to patients. Antidiabetic medications are free for retired people, severely ill people, and people with disabilities and at a very small cost for active patients.

### 2.1. Study Eligibility Criteria

The study population consisted of patients aged 75 years or older with a diagnosis of T2DM (ICD-10 codes E11 and E14 and subcodes) on 31 December 2016. Patients with any other type of diabetes were excluded from the analysis.

### 2.2. Study Variables

Variables collected were age, sex, time since diagnosis, body mass index (BMI), blood pressure, lipid profile, estimated Glomerular Filtration Rate (eGFR) using the Chronic Kidney Disease Epidemiology Collaboration (CKD-EPI) Equation, the urine albumin to creatinine ratio (UACR), and the most recent HbA1c value of the preceding 24 months. ICD-10 codes and data pertaining to other cardiovascular risk factors and chronic complications and comorbidities were also available and have been extensively described elsewhere [29]. To identify hypoglycemia, we used ICD-10 diagnostic code E16.2. Data on glucose-lowering medication were obtained from the CatSalut drug pharmacy invoice database.

### 2.3. Statistical Methods

Descriptive analysis consisted of summary statistics, mean and standard deviation for continuous variables, and percentages for categorical variables. Three subgroups of age were considered: 75–80, 81–85, and >85 years old. Glycemic control was stratified into four categories by HbA1c intervals (<6.5%, 6.5% to 6.9%, 7.0% to 7.9%, 8.0 to 8.9%, and ≥9%). According to renal function, patients were classified in three groups: normal/mild (eGFR ≥ 60 mL/min), moderate chronic renal failure (CRF) (eGFR 30–59 mL/min), and severe CRF (eGFR < 30 mL/min). Potential overtreatment was defined as having an HbA1c < 7% or HbA1c < 6.5% in patients aged 75 years or older treated with drugs associated with a high risk of hypoglycemia (insulin, sulfonylureas, or glinides). Steps of treatment and potential overtreatment were analyzed according to sex and age subgroups and the presence of other comorbidities: cardiovascular disease (CVD), chronic kidney disease (CKD), heart failure (HF), and severe CRF. Four antidiabetic steps of treatment—lifestyle modification only, non-insulin antidiabetic drug (NIAD) monotherapy, a combination of NIADS, and insulin alone or in combination with any NIAD—were considered. Antidiabetic drugs were classified in two groups: high risk of hypoglycemia (sulfonylureas, glinides, and insulin) and low risk of hypoglycemia (other NIADs).

### 2.4. Ethics Committee Approval

The study was approved by the Ethics Committee of the Primary Health Care University Research Institute (IDIAP) Jordi Gol (approval number: P17/015). In accordance with the Spanish regulations on observational studies, this retrospective study using pseudo-anonymized data did not require obtaining informed consent from the patient.

## 3. Results

By 31 December 2016, the SIDIAP database contained records from 138,374 patients with T2DM aged 75 years or older. The mean age was 82.6 (5.1) years, and 55% were women. In total, 123,515 (89.3%) patients had at least one HbA1c available. The mean HbA1c was 7.0% (Table 1); an HbA1c below 7.0% was present in 59.1% of patients and below 6.5% in 37.7% (Figure 1a). The degree of glycemic control stratified using HbA1c intervals (<6.5%, 6.5% to 6.9%, 7.0% to 7.9%, 8.0% to 8.9%, and ≥9%) for sex, age groups, and the considered comorbidities is shown in Figure 1a. Figure 1b shows the distribution of HbA1c intervals, displaying three subcategories in patients with HbA1c < 7%: treated with antidiabetic drugs with a high risk of hypoglycemia (26.7%), treated with low-risk antidiabetic drugs (16.9%), and treated only with lifestyle modification (15.6%). CKD was present in 51.9% of subjects, and severe CRF (eGFR < 30 mL/min) was present in 7.6% of 124,748 (90.15%) patients with at least one eGFR measurement available (Table 1).

### 3.1. Antidiabetic Treatment

Overall, 19.4% of patients were treated with only lifestyle changes, 43.5 with antidiabetics with a high risk of hypoglycemia (23.0% with insulin, 17.8% with sulfonylureas, 6.7% with glinides), and 37.6% with antidiabetic drugs with a low risk of hypoglycemia (Table 1). The distribution by treatment steps for each category of HbA1c showed that, in patients with HbA1c < 6.5%, 81.2% were treated with only lifestyle changes or monotherapy. Conversely, in patients with HbA1c ≥ 9%, 88.8% of them were treated with a NIAD combination or insulin (Figure 2a). The use of high-risk drugs increased as HbA1c increased: in patients with HbA1c < 6.5%, their use was 21.6%, while in those with HbA1c ≥ 9%, it was 86.9% (Figure 2b). Comparing the use of insulin in patients with HbA1c < 6.5% to those with HbA1c ≥ 9%, the percentages were 7.6% and 67.3%, respectively. These progressive increases were less marked for other high-risk antidiabetic drugs: 10.8% to 15.9% for sulfonylureas and 3.2% to 3.6% for glinides (Figure 2b).

Table 2 shows the steps of treatment and the use of antidiabetic drugs by age subgroups for each comorbidity. Patterns of use were quite similar among CKD, CVD, and HF patients in which insulin was more frequently prescribed (28.3%, 30.6%, and 35.7%, respectively) and, consequently, the percentage of high-risk antidiabetic drugs (49.0%, 49.8%, and 53.1%, respectively) (Table 1). These percentages decreased progressively when age increased, being lower in people older than 85 in all comorbidities but HF (Table 2). For instance: insulin in patients with severe CRF decreased from 59.8% in the 75–80-year-old group to 41.5% in the >85-year-old group. In a similar way, the use of high-risk antidiabetic drugs decreased from 72.2% to 58.2% in these age subgroups. Conversely, in patients with HF, the use of insulin increased from 30.6% to 36.1% in subjects older than 85 (Table 2).

### 3.2. Potential Overtreatment

Potential overtreatment in patients with an HbA1c < 7% using any high-risk antidiabetic drug was suspected in 26.6% of them (Figure 3), being higher in those treated with sulfonylurea or glinides (47.8% and 43.5%, respectively) than with insulin (28.1%). In Figure 1b, overtreatment for patients with an HbA1c < 7% is represented by the red area in each column, where its frequency can be visually compared to that of well-controlled patients not using any antidiabetic drugs (only lifestyle modification) or using only low-risk antidiabetic drugs. Using the threshold of an HbA1c < 6.5%, these figures were: 21.6% (any high-risk drug), 24.4% (sulfonylurea), 17.9% (glinides), and 12.3% (insulin).

No relevant differences in the sex and age groups were found. Potential overtreatment was more frequent in patients with CVD (31.6%), HF (34.8%), CKD (30.6%), and severe CRF (24.7%) for an HbA1c < 7% (Figure 1b and Figure 3). These figures for an HbA1c < 6.5% were 26.0%, 28.8%, 25.0%, and 20.7%, respectively (Figure 3).

Hypoglycemia as a health problem was registered in 1.2% of patients and was higher in persons treated with insulin (4.6% in monotherapy and 3.1% in association with NIADs) and glinides (1.5%), and it was lower with sulfonylureas (0.6%) and those treated with diet or low-risk antidiabetic drugs (both 0.3%) (Table 3). The frequency of registered hypoglycemia was slightly greater in women, 1.2 vs. 1.0% in men, and it increased with age (1.4% in older than 85). It also increased progressively as the HbA1c value increased, probably because of the greater use of high-risk antidiabetic drugs, especially insulin, in patients with an HbA1c above 8%: 2.5% between 8 and 8.9% and 2.7% above 9% (Figure 4). According to renal function, the frequency was greater in patients with severe CRF (2.5%) In relation to sulfonylureas, the frequency was quite similar in all subgroups and comorbidities (0.5 to 0.8%), being the greatest (0.8%) in three subgroups: patients older than 85, with an HbA1c < 6.5%, or with severe CRF. Finally, treatment with insulin monotherapy had a higher frequency than insulin in combination with NIADs, especially in patients older than 85, 5.0 vs. 3.8%, respectively, and in patients with moderate CRF (eGFR 30–59 mL/min), 5.6 vs. 3.2%, respectively.

## 4. Discussion

In this real-world population of patients with T2DM older than 75 years, 26.6% of those treated with antidiabetic drugs associated with a high risk of hypoglycemia had an HbA1c < 7% and, thus, they might be at potential risk of overtreatment. This proportion was higher in patients treated with sulfonylureas (47.8%) and glinides (43.5%) than those on insulin (28.1%). These results were lower using the threshold of an HbAc1 < 6.5%.

Nowadays, there is a consensus that there is a risk of overtreatment in patients older than 75 years or who are frail and for whom intensive treatment offers few clinical benefits but may cause important adverse effects, such as hypoglycemia. Therefore, these subjects are candidates for deprescribing [2,3,4,5,6,20,21,22,31]. In our database, glycemic goals were frequently achieved by older adults: nearly 60% had an HbA1c < 7%, 38% had an HbA1c < 6.5%, and the use of high-risk antidiabetic drugs such as insulin (23%), glinides (6.7%), and sulfonylureas (17.8%) was frequent. This unexpectedly high use of sulfonylureas is probably due to their initiation well before the introduction of other NIADs with a lower risk of hypoglycemia, but also because of the lower cost compared to other newer alternatives [32]. In recent years, there has been a progressive replacement of sulfonylureas with dipeptidyl peptidase 4 inhibitors (DPP4i) [26,33,34,35,36,37,38], especially, in older adults. As has been published previously, in our database, 17.8% of T2DM patients older than 75 were still treated with sulfonylureas, while 16.1% received a DPP4i in the same study period [30]. However, in several studies around the world, the prescription of DPP4i progressively outperformed sulfonylureas in older adults [26,33,37,38]. This could explain the progressive decrease in sulfonylureas use observed in our database from 2007 to 2018 [28].

In the last decade, different studies have shown that overtreatment, defined as having an HbA1c < 7% in patients treated with insulin, sulfonylureas, or glinides, is a frequent issue in older adults, ranging from 26% to 62% [12,13,14,15,16,17,18,24,25,26]. These wide differences could be related to the different types of populations or prescribing habits, but also to the age threshold used in each study. Most of them included patients older than 65 [12,13,14,15,16], but others were older than 70 [16,17,25], or even over 75 [18,26]. In our study, we used the threshold of 75, following the recommendations of European and national primary care guidelines [3,4]. In one study from the US including 42,669 T2DM patients older than 75, potential overtreatment was 26% [18], while in a small study in Italy with 387 patients, it was 62% [24]. Results in people over 70 were 38.8% in Holland [16] and 29.9% in the UK [17]. In our study, in people over 75, slight differences in sex or age groups were found, but they were greater in the presence of the studied comorbidities, from 30.6% with CKD to 34.8% in patients with HF. The higher risk of overtreatment in patients treated with sulfonylureas and glinides in comparison to insulin is related to the fact that the mean HbA1c is higher in patients treated with insulin.

Primary care professionals must consider discussing deintensification as part of routine care to determine the goals of care and make shared decisions according to patient health status, glycemic control, and patient preferences and values [19,20,21,22]. Deintensification may include deprescribing antidiabetics and other medications or reducing home blood glucose monitoring and diabetes-specific assessments that no longer improve quality of life and life expectancy in older adults. When deprescribing, professionals may not only consider the reduction of insulin doses and the simplification of complex insulin regimes but also stopping sulfonylurea treatment and starting an antidiabetic that does not induce hypoglycemia [3,4,5,6,20,21,22]. In addition to advising against intensive glycemic control in these patients, it would be necessary to introduce the practice of deprescribing if a person’s level of HbA1c is below 7.0%, either lowering doses, switching to a safer medication, or stopping medications [19,20,21,22]. A metanalysis of randomized clinical trials confirmed that deprescribing is safe in these patients [23]. Despite that, deprescribing is uncommon in clinical practice [24,25,26], even in individuals with limited life expectancy [17,25].

Surprisingly, hypoglycemia was registered in only 1596 patients (1.2%), out of which 73.2% were on insulin, 9% on sulfonylureas, and 8.7% on glinides. Its frequency was slightly higher in women and increased with age, as well as with decreased kidney function and greater levels of HbA1c. This low registration has been previously reported by other authors [39,40] and probably reflects the limitations of the EHR in primary care: only severe or very frequent events are probably recorded. In any case, in our EHR, they remain active as a reminder/alert for health care professionals. Moreover, the higher frequency observed in patients treated with glinides and sulfonylureas probably represents the switch from sulfonylureas to glinides due to previous episodes of hypoglycemia that remain recorded as a health problem. It might be surprising that poorer glycemic control was associated with an increased risk of hypoglycemia. However, this is probably attributable to the greater use of high-risk antidiabetic drugs, especially insulin, in these patients. In fact, observational studies have yielded conflicting results; some of them have found an increased risk of hypoglycemia at lower HbA1c levels [10], whereas others have shown an increased risk of hypoglycemia at both lower and higher HbA1c levels [8,11]. For instance, Lipska et al. found that severe hypoglycemia was common among patients with type 2 diabetes across all levels of glycemic control, and the risk tended to be higher in patients with either near-normal or very poor glycemic control [8]. Additionally, Ling et al. found that sulfonylurea and insulin use were more relevant predictors of severe hypoglycemia and death than glucose levels [11].

Mild-to-moderate events are usually underestimated in database studies since they can only be detected if they are recognized and reported by patients to their providers and are subsequently recorded within the clinical notes and coded as a health problem. For instance, in a US study, ascertainment of events using natural language processing (NLP) that identifies whether hypoglycemia is mentioned in clinical notes increased the capture of non-serious events more than 20-fold in comparison to the structured data recorded [39]. This study also showed that hypoglycemia was highly prevalent, with over 10% of patients per year experiencing at least one event that was documented [39]. Likewise, in another US study using the same methodology, in 317,399 T2DM patients, the prevalence of recorded hypoglycemia was 4%, while, according to the NLP, it was 8% [40]. These figures, even higher than ours, seem to be also lower than expected, as 55% of T2DM patients in that study were using insulin [40].

Our study has several limitations. Since most of the selected clinical conditions were based on those diagnoses recorded in the database, under-registration and misclassification cannot be ruled out. Moreover, we were not able to identify frailty in our database and had to assume that an age older than 75 years is a high-risk category in all cases. Furthermore, HbA1c is not always contemporary with the active medication at the study cutoff point. We assume that the risk probably persisted until the end of the study period. Finally, the main limitation is related to the low registration of hypoglycemia as a health problem in our EHR, suggesting that only the most severe or repeated cases were recorded. In addition, other acute or chronic complications were not considered in our analysis as they have been published previously [29].

The strengths of our study include a population-based design; the use of a primary care database with a large number of subjects; and, unlike other population-based studies, the fact that HbA1c and eGFR values were available in almost 90% of cases.

## 5. Conclusions

One in four older adults with T2DM treated with antidiabetic drugs associated with a high risk of hypoglycemia might be at risk of overtreatment. This risk is higher in those treated with sulfonylureas or glinides than with insulin. Chronic kidney disease is a frequent comorbidity among this subpopulation and should be considered in treatment choices. Less stringent glycemic goals, the deintensification of the treatment, and/or changes to the antidiabetic treatment using other drugs with a lower risk of hypoglycemia should be considered in the high-risk population, avoiding drugs that are contraindicated in patients with chronic renal failure. Finally, recording hypoglycemia as a health problem in the EHR was low, although it was more frequent in patients receiving insulin.

## Figures and Tables

**Figure 1 jcm-11-05134-f001:**
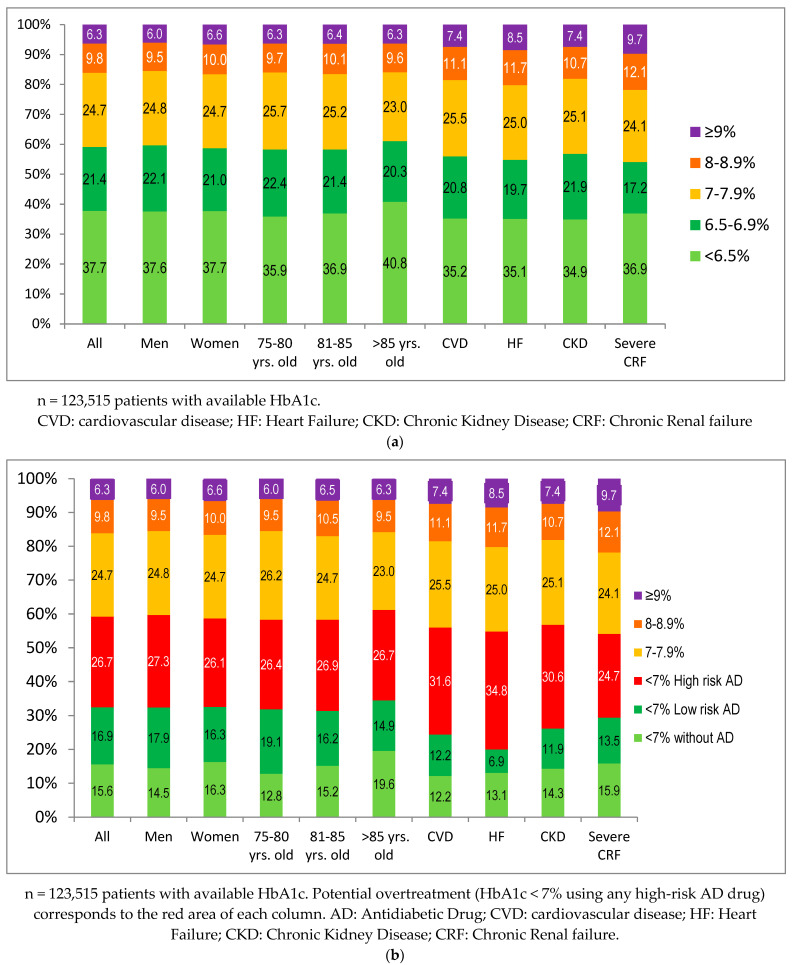
Degree of glycemic control. (**a**) Degree of glycemic control (HbA1c categories) by sex, age subgroups, and comorbidities; (**b**) Degree of glycemic control (HbA1c categories) by sex, age subgroups, and comorbidities considering the use of antidiabetic drugs in patients with HbA1c < 7%.

**Figure 2 jcm-11-05134-f002:**
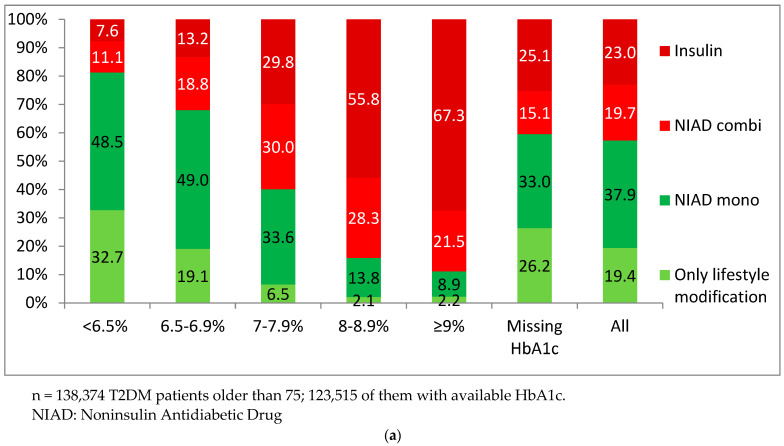
Antidiabetic treatment. (**a**) Steps of antidiabetic treatment by HbA1c categories; (**b**) Use of high-risk antidiabetic drugs by HbA1c categories.

**Figure 3 jcm-11-05134-f003:**
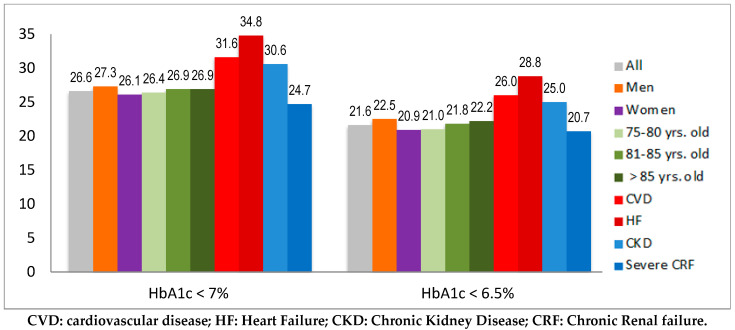
Potential overtreatment according to an HbA1c < 7.0% or an HbA1c < 6.5% criteria by sex and age subgroups and comorbidities.

**Figure 4 jcm-11-05134-f004:**
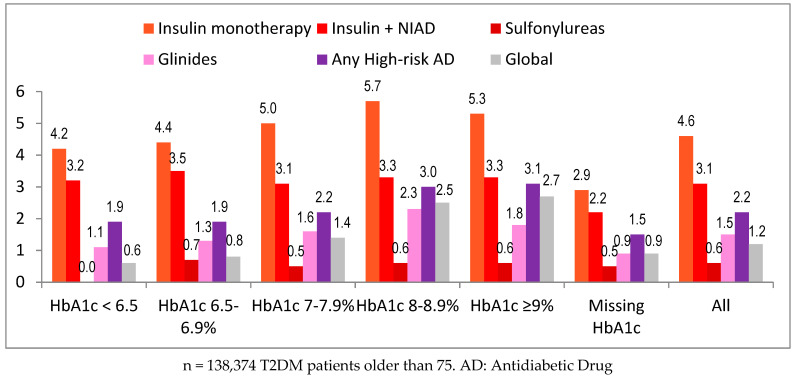
Registered hypoglycemia according to the use of high-risked antidiabetic drugs and HbA1c categories. *n* = 138,374, T2DM patients older than 75. AD: antidiabetic drug.

**Table 1 jcm-11-05134-t001:** Clinical characteristics of patients by sex and age subgroups and the presence of comorbidities.

Variables	Overall *n* = 138,374	Men *n* = 61,449	Women *n* = 76,925	75–80 Years Old *n* = 48,951	81–85 Years Old *n* = 47,278	>85 Years Old *n* = 42,145	CVD *n* = 44,668	HF *n* = 17,986	CKD *n* = 71,848	Severe CRF *n* = 5272
Mean age, years (SD)	82.6 (5.1)	81.9 (4.8)	83.2 (5.3)	77.5 (1.5)	82.9 (1.1)	88.9 (3.1)	83.1 (5.1)	84.2 (5.3)	83.5 (5.2)	84.7 (5.4)
Gender (female), %	55.6	0	100	49.2	55.1	63.5	44.3	59.6	57.1	59.8
Mean diabetes duration, years (SD)	11.2 (6.8)	11.1 (6.6)	11.3 (7.0)	10.7 (6.4)	11.4 (6.8)	11.7 (7.1)	12.0 (7.1)	11.7 (7.1)	11.8 (6.9)	12.9 (7.2)
Mean HbA1c, % (SD) (*n* = 123,515)	7.0 (1.1)	7.0 (1.1)	7.0 (1.1)	7.0 (1.1)	7.0 (1.1)	7.0 (1.1)	7.1 (1.1)	7.1 (1.2)	7.1 (1.1)	7.1 (1.2)
Current smoker, %	4.8	9.1	1.3	6.9	4.5	2.6	5.6	3.3	4.5	3.7
Mean BMI, kg/m^2^ (SD) (*n* = 117,212)	28.9 (3.9)	28.2 (3.9)	29.5 (5.2)	29.6 (4.8)	28.9(4.7)	27.9 (4.5)	28.5 (4.5)	30.0 (5.4)	29.1 (4.8)	29.4 (5.2)
Obesity (BMI > 30 kg/m^2^), % (*n* = 117,212)	36.1	29.2	41.9	41.1	36.7	28.5	33.0	16.9	37.9	41.2
Hypertension, %	84.4	80.3	87.7	82.5	84.9	86.1	85.9	89.2	89.5	91.6
Hyperlipidemia, %	60.4	56.3	63.7	63.8	61.1	55.7	63.0	60.4	61.8	60.9
CVD, %	32.3	40.5	25.7	29.2	32.6	35.6	-	49.4	37.1	45.3
HF, %	13.0	11.8	13.9	8.9	12.6	18.2	19.9	-	17.8	33.2
Retinopathy, %	15.8	15.0	16.4	15.4	16.2	15.8	18.8	19.6	17.8	22.1
Neuropathy, %	10.0	8.4	11.3	10.5	10.45	8.8	11.4	11.8	10.8	9.0
CKD (eGFR < 60 mL/min/1.73 m^2^ or albuminuria > 30 mg/g), %	51.9	50.2	53.3	42.1	53.8	62.5	59.7	71.1	-	-
Severe chronic renal failure (eGFR < 30 mL/min/1.73 m^2^), % (*n* = 124,748)	7.6	7.0	8.2	4.6	7.1	11.8	10.7	19.3	-	-
**Antidiabetic Treatment**										
Only lifestyle modification, %	19.4	18.0	20.6	15.6	18.5	25.0	17.0	18.9	17.9	20.1
Non-insulin antidiabetic drug monotherapy, %	37.9	38.6	37.4	38.4	37.8	37.5	34.3	31.8	35.2	25.7
Non-insulin antidiabetic drug combination, %	19.7	21.7	18.0	23.1	20.0	15.3	18	13.7	18.6	7.8
Insulin (alone or in combination), %	23.0	21.7	24.0	22.9	23.7	22.3	30.6	35.7	28.3	46.5
Sulfonylureas, %	17.8	18.8	17.8	19.9	17.8	15.3	16.4	13.8	16.9	17.2
Glinides, %	6.7	6.8	6.6	6.0	6.9	7.4	7.6	8.9	8.6	4.6
High-risk antidiabetics, %	43.5	43.3	43.6	44.5	44.1	41.6	49.8	53.1	49.0	66.0
Only low-risk antidiabetics, %	37.6	38.7	35.8	39.9	37.4	33.4	33.2	28.0	33.1	13.1

CVD: cardiovascular disease; HF: heart failure; CKD: chronic kidney disease; CRF: chronic renal failure; eGFR: estimated Glomerular Filtration Rate. High-risk antidiabetic drugs: sulfonylureas, glinides, insulin; low-risk ADs: metformin, SGLT-2 inhibitors, GLP-1 receptor agonists, DPP4 inhibitors, alfa-glucosidase inhibitors, pioglitazone.

**Table 2 jcm-11-05134-t002:** Steps of treatment and use of antidiabetic drugs by age subgroups and the presence of comorbidities.

	Overall *n* = 138,374	CVD *n* = 39,928			HF *n* = 17,986			CKD *n* = 71,848			Severe CRF (eGFR < 30 mL/min) *n* = 5272		
		75–80 yrs. *n* = 17,504	81–85 yrs. *n* = 15,766	>85 yrs. *n* = 12,398	75–80 yrs. *n* = 5506	81–85 yrs. *n* = 5994	>85 yrs. *n* = 6486	75–80 yrs. *n* = 25,716	81–85 yrs. *n* = 24,219	>85 yrs. *n* = 21,193	75–80 yrs. *n* = 1379	81–85 yrs. *n* = 1703	>85 yrs. *n* = 2190
Only lifestyle modification, %	19.4	13.1	15.8	24.1	13.0	17.3	25.3	13.4	17.0	24.1	14.0	17.4	27.9
Non-insulin antidiabetic drug monotherapy, %	37.9	33.6	34.6	35.1	30.4	31.6	33.1	34.2	35.2	36.3	20.6	21.0	25.1
Non-insulin antidiabetic drug combination, %	19.7	21.0	18.0	13.9	16.0	14.3	11.1	22.2	18.7	14.4	5.6	4.6	5.6
Insulin (alone or in combination), %	23.0	32.3	31.6	26.9	30.6	33.3	36.1	30.2	29.0	25.2	59.8	57.0	41.5
Sulfonylureas, %	17.8	18.3	15.9	14.3	15.3	13.9	12.3	18.8	16.8	14.7	2.8	4.1	6.3
Glinides, %	6.7	6.9	7.8	8.2	8.1	9.0	9.4	8.0	8.8	9.1	17.0	18.6	16.3
Any high-risk antidiabetic drug, %	43.5	52.3	50.4	45.5	57.9	54.2	47.9	51.8	49.6	45.0	72.2	71.1	58.2
Any low-risk antidiabetic drug, %	37.6	34.6	33.8	31.3	29.1	28.5	26.8	34.8	33.4	30.9	13.8	11.5	13.9

CVD: cardiovascular disease; HF: heart failure; CKD: chronic kidney disease; CRF: chronic renal failure; eGFR: estimated Glomerular Filtration Rate. High-risk antidiabetic drugs: sulfonylureas, glinides, insulin; low-risk ADs: metformin, SGLT-2 inhibitors, GLP-1 receptor agonists, DPP4 inhibitors, alfa-glucosidase inhibitors, pioglitazone.

**Table 3 jcm-11-05134-t003:** Frequency of registered hypoglycemia in patients treated with drugs associated with a higher risk of hypoglycemia (insulin, sulfonylureas, or glinides) and treatment steps according to sex and age subgroups, the categories of HbA1c, and estimated Glomerular Filtration Rate.

	Overall	Women	Men	75–80 yrs.	81–85 yrs.	>85 yrs.	HbA1c < 6.5%	HbA1c 6.5–6.9%	HbA1c 7–7.9%	HbA1c 8–8.9%	HbA1c ≥ 9%	Missing HbA1c	eGFR > 60 mL/min	eGFR 30–59 mL/min	eGFR < 30 mL/min
	*n* = 13,8374	*n* = 76,925	*n* = 61,449	*n* = 59,440	*n* = 44,314	*n* = 34,620	*n* = 46,554	*n* = 21,705	*n* = 31,262	*n* = 12,713	*n* = 6858	*n* = 19,282	*n* = 74,817	*n* = 43,292	*n* = 5138
All patients, *n* (%)	1596 (1.2)	958 (1.2)	638 (1.0)	466 (1.0)	424 (1.2)	589 (1.4)	301 (0.6)	178 (0.8)	446 (1.4)	321 (2.5)	186 (2.7)	164 (0.9)	682 (0.9)	685 (1.6)	134 (2.5)
Sulfonylurea, *n* (%)	150 (0.6)	90 (0.7)	60 (0.5)	53 (0.5)	38 (0.6)	49 (0.8)	43 (0.8)	24 (0.7)	38 (0.5)	21 (0.6)	9 (0.6)	15 (0.5)	89 (0.6)	51 (0.7)	2 (0.8)
Glinides, *n* (%)	139 (1.5)	82 (1.6)	57 (1.4)	27 (0.9)	41 (1.6)	63 (2.0)	19 (1.1)	14 (1.3)	42 (1.6)	37 (2.3)	15 (1.8)	12 (0.9)	44 (1.3)	66 (1.6)	20 (2.2)
Insulin, *n* (%)	1168 (3.7)	720 (3.9)	448 (3.4)	350 (3.1)	312 (3.6)	414 (4.4)	132 (3.7)	110 (3.9)	350 (3.8)	289 (4.1)	178 (3.9)	109 (2.5)	481 (3.5)	515 (3.9)	106 (4.2)
Any high-risk AD, *n* (%)	1343 (2.2)	820 (2.4)	523 (2.0)	405 (1.9)	360 (2.2)	478 (2.7)	192 (1.9)	140 (1.9)	397 (2.2)	310 (3.0)	182 (3.1)	122 (1.5)	569 (1.9)	581 (2.7)	120 (3.4)
Only low risk AD, *n* (%)	253 (0.3)	138 (0.3)	115 (0.3)	61 (0.2)	64 (0.3)	111 (0.5)	109 (0.3)	38 (0.3)	49 (0.4)	11 (0.5)	4 (0.4)	42 (0.4)	113 (0.2)	104 (0.5)	14 (0.8)
Lifestyle modification, *n* (%)	80 (0.3)	51 (0.3)	29 (0.3)	11 (0.1)	25 0.4)	40 (0.4)	42 (0.3)	10 (0.2)	6 (0.3)	4 (1.5)	0 (0.0)	18 (0.4)	26 (0.2)	33 (0.4)	9 (0.8)
NIAD monotherapy, *n* (%)	198 (0.4)	105 (0.4)	93 (0.4)	53 (0.3)	46 (0.3)	88 (0.6)	93 (0.4)	33 (0.3)	36 (0.3)	8 (0.5)	2 (0.3)	26 (0.4)	92 (0.3)	77 (0.5)	16 (1.3)
NIAD combination, *n* (%)	150 (0.6)	82 (0.6)	68 (0.5)	52 (0.5)	41 (0.6)	47 (0.7)	34 (0.7)	25 (0.6)	54 (0.6)	20 (0.6)	6 (0.4)	11 (0.3)	83 (0.5)	60 (0.8)	3 (1.1)
Insulin monotherapy, *n* (%)	548 (4.6)	332 (4.7)	216 (4.5)	141 (4.4)	137 (4.2)	231 (5.0)	75 (4.2)	52 (4.4)	161 (5.0)	127 (5.7)	77 (5.3)	56 (2.9)	155 (4.3)	281 (5.6)	80 (4.3)
Insulin + NIAD, *n* (%)	620 (3.1)	388 (3.4)	232 (2.7)	209 (2.6)	175 (3.2)	183 (3.8)	57 (3.2)	58 (3.5)	189 (3.1)	162 (3.3)	101 (3.2)	53 (2.2)	326 (3.2)	234 (3.2)	26 (3.1)

AD: antidiabetic drug; eGFR: estimated Glomerular Filtration Rate; NIAD: non-insulin antidiabetic drugs. High-risk ADs: sulfonylureas, glinides, insulin; low-risk ADs: metformin, SGLT-2 inhibitors, GLP-1 receptor agonists, DPP4 inhibitors, alfa-glucosidase inhibitors, pioglitazone.

## Data Availability

The data that support the findings of this study are available in the SIDIAP database (System for the Development of Research in Primary Care). Restrictions apply to the availability of these data, which were used under license for this study.

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
