# Peer review of "Potential Risk of Overtreatment in Patients with Type 2 Diabetes Aged 75 Years or Older: Data from a Population Database in Catalonia, Spain"

_jcm, 2022, doi:10.3390/jcm11175134_

Round 1

Reviewer 1 Report

POTENTIAL RISK OF OVERTREATMENT IN PATIENTSWITH TYPE 2 DIABETES AGED 75 YEARS OR OLDER: DATA FROM A POPULATION DATABASE IN CATALONIA, SPAIN

The article deals with a topic of relevance, but in reality it is necessary to have more cases among other topics. This work would be a preliminary work that accounts for the responses to different treatments in type 2 diabetes, in older adults.

On the other hand the first figures need to add information in order to facilitate interpretation.

Author Response

Reviewer 1

Comments and Suggestions for Authors

POTENTIAL RISK OF OVERTREATMENT IN PATIENTS WITH TYPE 2 DIABETES AGED 75 YEARS OR OLDER: DATA FROM A POPULATION DATABASE IN CATALONIA, SPAIN

  1. The article deals with a topic of relevance, but in reality it is necessary to have more cases among other topics. This work would be a preliminary work that accounts for the responses to different treatments in type 2 diabetes, in older adults.

We thank the reviewer for this comment. We agree that in older people all drugs can produce adverse effects more frequently than in younger subjects. However, due to the intrinsic limitations of the design, in a cross-sectional study it is not possible to correlate these events with a given antidiabetic drug. A previous publication from our group provided details on the prevalence of other acute and chronic complications (reference 29) in type 2 diabetes patients aged over 75 years.

Following the suggestion of the Reviewer, in the paragraph on limitations of the discussion, we have now added a sentence commenting the risk of other complications that are not considered in our publication (Page 11, line 35 before the strengths):

“Besides, other acute or chronic complications have not been considered in our analysis as they have been published previously (29).”

  1. On the other hand, the first figures need to add information in order to facilitate interpretation.

After a careful revision of these figures, we have modified the wording of some titles, and we moved some information to footnotes. We hope that these changes have improved the understanding of the reader.

Reviewer 2 Report

The study is on an important topic

Well conducted and presented.

Retrospective nature of the study and abscence on the data on number of hypoglycemic episodes and severe hypoglycemic episodes is a significant limitation.

Author Response

Reviewer 2

Comments and Suggestions for Authors

The study is on an important topic

Well conducted and presented.

Retrospective nature of the study and absence on the data on number of hypoglycemic episodes and severe hypoglycemic episodes is a significant limitation.

We thank the Reviewer for his remark on this limitation. As it is stated in the discussion, “the main limitation is related to the low registration of hypoglycemia as a health problem in our EHR, suggesting that only most severe or repeated cases were recorded.”  However, assessing the association between overtreatment and risk of hypoglycemia was neither the main objective of the study, nor was it designed to do so. The main objective of the study was to identify the risk of overtreatment in this population and, as a secondary objective, to determine the percentage of hypoglycemia recorded for each subgroup of patients.

Reviewer 3 Report

The study is focused on diabetes overtreatment in the elderly, which is an essential issue . The study is well designed and provides helpful information. Just a minor comment, some of the results are repeated in different figures, confusing the reader.  

Author Response

Reviewer 3

The study is focused on diabetes overtreatment in the elderly, which is an essential issue . The study is well designed and provides helpful information. Just a minor comment, some of the results are repeated in different figures, confusing the reader. 

We thank the Reviewer on this comment we did changes in the figures.

Reviewer 4 Report

The authors investigated the management status of aged people with T2DM using electronic health records. The aim of the study is valuable and the findings are also interesting.  But some points should be improved. 

1. About overtreatment 

The authors suggested hypoglycemia as the significant side effect of overtreatment. However, Figure 4 shows similar hypoglycemic events among the study groups. The highest risk group is the HbA1c 8~8.9% group.

2. Other complications 

Hypoglycemia is one of the various complications of T2DM. Thus, other complications should be considered when we think about over- or under-treatment. Thus, the authors should investigate other major complications of T2DM according to HbA1c level. 

Author Response

Reviewer 4

The authors investigated the management status of aged people with T2DM using electronic health records. The aim of the study is valuable, and the findings are also interesting.  But some points should be improved.

  1. About overtreatment

The authors suggested hypoglycemia as the significant side effect of overtreatment. However, Figure 4 shows similar hypoglycemic events among the study groups. The highest risk group is the HbA1c 8~8.9% group.

We highly appreciate this Reviewer’s comment. Actually, the risk of hypoglycemia is the most relevant adverse effect in the definition of overtreatment. As it is stated in the discussion, the main limitation of our study is related to the low recording of hypoglycemia as a health problem in the EHR [hypoglycemia was recorded as a diagnostic code in only 1,596 patients (1.2%)]. Following the Reviewer’s appraisal, we agree that ongoing frequency of hypoglycemic episodes rising progressively with HbA1c values seems to be counterintuitive; on one hand, this might probably be due to the greater use of high-risk antidiabetic drugs in these patients, especially insulin. On the other hand, after checking further literature, although there is a discordancy between studies, the finding that higher HbA1c might be associated to an increased risk of hypoglycemia is not surprising. We would like to point out the conclusion of the Lipska et al study, in which the association between HbA1c level and self-reported severe hypoglycemia in patients with type 2 diabetes was examined: “Severe hypoglycemia was common among patients with type 2 diabetes across all levels of glycemic control. Risk tended to be higher in patients with either near-normal glycemia or very poor glycemic control” (8). In fact, observational studies have yielded conflicting results: some of them have indicated an increased risk of hypoglycemia at lower HbA1c levels (10), whereas others have shown increased risk of hypoglycemia at both lower and higher HbA1c levels (8,11). Additionally, Ling et al found that sulfonylurea and insulin use were more relevant predictors of severe hypoglycemia and death than glucose levels (11). Nevertheless, the real value about interpreting these data might be to show that higher hypoglycemia rates with insulin, glinides and SU can be found in real life irrespective of the HbA1c level.

Following the suggestion of the Reviewer, we have added these comments in the part of the discussion dealing with hypoglycemia (Page 11, line 18 (before “Mild-to-moderate events are usually ….):

“It might be surprising that poorer glycemic control was associated with an increased risk of hypoglycemia. However, this is probably attributable to the greater use of high-risk antidiabetic drugs, especially insulin, in these patients. In fact, observational studies have yielded conflicting results; some of them have found an increased risk of hypoglycemia at lower HbA1c levels (10), whereas others have shown increased risk of hypoglycemia at both lower and higher HbA1c levels (8,11). For instance, Lipska et al found that severe hypoglycemia was common among patients with type 2 diabetes across all levels of glycemic control and the risk tended to be higher in patients with either near-normal or very poor glycemic control (8). Additionally, Ling et al found that sulfonylurea and insulin use were more relevant predictors of severe hypoglycemia and death than glucose levels (11).”

  1. Other complications

Hypoglycemia is one of the various complications of T2DM. Thus, other complications should be considered when we think about over- or under-treatment. Thus, the authors should investigate other major complications of T2DM according to HbA1c level.

We also thank the Reviewer for this comment. Of course, in older people all drugs can produce adverse effects more frequently than in younger subjects. However, due to the intrinsic limitations of the design, in a cross-sectional study it is impossible to correlate these events with a given antidiabetic drug. A previous publication from our group already provided details on the prevalence of other acute and chronic complications and comorbidities in type 2 diabetes patients aged over 75 (29). For instance, Hypertension, CKD, CVD, CHF, retinopathy and urinary tract infections were more frequent , in people aged > 75, while obesity, liver disease, genital infections and neuropathy were less prevalent. Finally, no differences were observed in the prevalence of dyslipidemia and pancreatitis (29). These differences can be seen in figure 2 of the original publication (enclosed below):

Moreover, the term overtreatment is usually limited to hypoglycemia in most publications. In fact, the Standards of Care issued by the American Diabetes Association include specific comments on the risk of hypoglycemia in relation to strict glycemic control: “Tight glycemic control in older adults with multiple medical conditions is considered overtreatment and is associated with an increased risk of hypoglycemia” (5). In the same line, the Expert Consensus Statement on the Management of Older Adults with Type 2, published in 2021, focused on the risk of hypoglycemia: “As a person with diabetes gets older, simplification, switching or de-escalation of the therapeutic regimen may be necessary, depending on their level of frailty and HbA1c levels. Consideration should be given, in particular, to de-escalation of therapies that may induce hypoglycaemia, such as sulphonylureas and shorter-acting insulins.” (6).

Finally, given the high evidence about the major importance of hypoglycemia among older people (with highest incidence of cardiovascular events, falls, neurocognitive decline, and even increased mortality rate) [1–4], we considered appropriate to focus the study on the identification of overtreatment criteria to detect the indirect risk of hypoglycemia among aged people.

[1]       Lega IC, Campitelli MA, Austin PC, Na Y, Zahedi A, Leung F, et al. Potential diabetes overtreatment and risk of adverse events among older adults in Ontario: a population-based study. Diabetologia 2021;64:1093–102. https://doi.org/10.1007/s00125-020-05370-7.

[2]       Whitmer RA, Karter AJ, Yaffe K, Quesenberry CP, Selby J v. Hypoglycemic episodes and risk of dementia in older patients with type 2 diabetes mellitus. JAMA - Journal of the American Medical Association 2009;301:1565–72. https://doi.org/10.1001/jama.2009.460.

[3]       Goto A, Arah OA, Goto M, Terauchi Y, Noda M. Severe hypoglycaemia and cardiovascular disease: Systematic review and meta-analysis with bias analysis. BMJ (Online) 2013;347. https://doi.org/10.1136/bmj.f4533.

[4]       Bonds DE, Miller ME, Bergenstal RM, Buse JB, Byington RP, Cutler JA, et al. The association between symptomatic, severe hypoglycaemia and mortality in type 2 diabetes: Retrospective epidemiological analysis of the ACCORD study. BMJ (Online) 2010;340:137. https://doi.org/10.1136/bmj.b4909.

Following the suggestion of the Reviewer, in the paragraph on the limitations of the discussion, we have added a sentence mentioning the risk of other complications that are not considered in our publication (Page 11, last line 35 before the strengths):

“Besides, other acute or chronic complications have not been considered in our analysis as they have been published previously (29).”

Round 2

Reviewer 4 Report

The authors have satisfactorily responded to all my questions.